# Facilitators and Barriers of Artificial Intelligence Applications in Rehabilitation: A Mixed-Method Approach

**DOI:** 10.3390/ijerph192315919

**Published:** 2022-11-29

**Authors:** Mashael Alsobhi, Harpreet Singh Sachdev, Mohamed Faisal Chevidikunnan, Reem Basuodan, Dhanesh Kumar K U, Fayaz Khan

**Affiliations:** 1Department of Physical Therapy, Faculty of Medical Rehabilitation Sciences, King Abdulaziz University, Jeddah 22252, Saudi Arabia; 2Department of Neurology, All India Institute of Medical Sciences, New Delhi 110029, India; 3Department of Rehabilitation Sciences, College of Health and Rehabilitation Sciences, Princess Nourah bint Abdulrahman University, P.O. Box 84428, Riyadh 11671, Saudi Arabia; 4Nitte Institute of Physiotherapy, Nitte University, Deralaktte, Mangalore 575022, India

**Keywords:** Artificial intelligence, physical therapist, intention to use, perceived barriers, clinical decision support

## Abstract

Artificial intelligence (AI) has been used in physical therapy diagnosis and management for various impairments. Physical therapists (PTs) need to be able to utilize the latest innovative treatment techniques to improve the quality of care. The study aimed to describe PTs’ views on AI and investigate multiple factors as indicators of AI knowledge, attitude, and adoption among PTs. Moreover, the study aimed to identify the barriers to using AI in rehabilitation. Two hundred and thirty-six PTs participated voluntarily in the study. A concurrent mixed-method design was used to document PTs’ opinions regarding AI deployment in rehabilitation. A self-administered survey consisting of several aspects, including demographic, knowledge, uses, advantages, impacts, and barriers limiting AI utilization in rehabilitation, was used. A total of 63.3% of PTs reported that they had not experienced any kind of AI applications at work. The major factors predicting a higher level of AI knowledge among PTs were being a non-academic worker (OR = 1.77 [95% CI; 1.01 to 3.12], *p* = 0.04), being a senior PT (OR = 2.44, [95%CI: 1.40 to 4.22], *p* = 0.002), and having a Master/Doctorate degree (OR = 1.97, [95%CI: 1.11 to 3.50], *p* = 0.02). However, the cost and resources of AI were the major reported barriers to adopting AI-based technologies. The study highlighted a remarkable dearth of AI knowledge among PTs. AI and advanced knowledge in technology need to be urgently transferred to PTs.

## 1. Introduction

One of the physical therapists’ (PTs) responsibilities is to perform physical rehabilitation assessment to design an appropriate clinical plan of care for patients with physical disorders such as stroke [1] and the anterior cruciate ligament [2]. Indeed, this initial process depends on PT’s experience, and sometimes it is not easily performed due to time constraints or limited availability of the human workforce [3]. Artificial intelligence (AI) has been expanding very rapidly in rehabilitation settings. Continuing research persists in developing smart machines to help assist therapists and monitor diseases [4].

AI is an algorithm process that has been used in healthcare and rehabilitation fields to generate decision-making and facilitate patient care services [5,6]. Moreover, it can be used to produce clinical predictions based on patients’ input data [7]. At present, AI-advanced technologies are influencing the healthcare system in every aspect, including educational, social, economical, and legal aspects that may have an impact on one’s life [8]. In medical practices, AI-based applications process large healthcare data sets using sophisticated algorithms to provide understandings that help clinicians in their medical management. Moreover, these advanced technologies are designed to learn and self-correct to improve output accuracy based on feedback. However, AI technologies are used to improve patients’ quality of care. They also aim to provide healthcare practitioners with recent medical information that is driven by various scientific sources such as articles, books, or clinical practices [9]. Moreover, numerous studies mentioned the advantages of using AI in clinical practices, and one of these advantages is minimizing diagnostic and therapeutic medical errors that are expected to occur in daily human practices [6,7,8].

Given that focal changes could arise by implanting AI in medical practices, more research was conducted to investigate AI knowledge and attitudes of healthcare practitioners in various specialties [10,11,12,13]. In acute care trauma and emergency surgeries, researchers stated that it is crucial to promote AI utilization among emergency surgeons by providing training courses, as 63% of surgeons have not used robotic systems in their clinical settings [14]. Another study investigated ophthalmologists’ perceptions regarding AI, and a positive opinion was reported among ophthalmologists. However, researchers enforced that it is essential to further investigate the effective approaches to implementing AI in clinical practices [15]. Moreover, in Australia and New Zealand, an electronic survey study was conducted to investigate the opinions of various healthcare providers on automation technologies. Researchers concluded the urgency of discovering healthcare professionals’ and patients’ perceptions of AI to successfully implement AI in healthcare [10]. However, factors affecting AI adoption and acceptance have not been studied, especially in physical therapy.

Robotics and AI tools have a remarkable impact on healthcare and rehabilitation care delivery services. Recent studies reported that higher accuracy and faster medical diagnosis and predictions could be obtained from employing AI applications that improve patients’ outcomes [16,17,18]. For example, a previous study showed the accuracy of deep learning, which is a part of AI functions, in determining skin cancer more accurately and efficiently than human diagnosis [19]. Moreover, another study documented that AI has been used to detect breast cancer, and successful implementation was found in terms of minimizing human errors [20]. However, ethical concerns and technology trust issues are still a dilemma in AI medical deployment [21].

In PT practices, AI systems can be used to train patients and monitor progress either by using virtual (informatics) or physical (robotics) AI concepts [22]. In a study that was conducted by Wei et al. (2019) [23], researchers studied the potentiality of a virtual PT system to improve the balance and mobility of patients with Parkinson’s disease remotely. It was found that AI technology had high accuracy level in remote training and detecting patients’ management processes. Moreover, AI can be used to monitor and enhance patient adherence to therapeutic exercises in various musculoskeletal cases, such as neck or back pain. In a study that was conducted by Lo et al. (2018) [24], researchers found that using AI-enabled mobile applications with patients who had neck and back pain was beneficial in terms of increasing therapeutic exercise adherence.

Moreover, supervised machine learning was studied to investigate the ability of AI-enabled technology to monitor patients’ exercise adherence at home. A study conducted in 2018 by Burns et al. [25] demonstrated the technical feasibility of supervised machine learning to track adherence to rotator cuff exercise regimens at home among healthy individuals, which improves patients’ healthcare outcomes. AI interventions have been developed not only as cost-effective procedures but also to enhance the quality of care. Researchers compared conventionally and AI digital sessions after total knee replacement (TKR) surgery among patients with knee osteoarthritis. The digital sessions employed 3D movement quantification to detect patients’ motion via a phone application and a web-based site. The study concluded that digital intervention for a home program after TKR surgery reduced the therapists’ workload and maximized patients’ outcomes [26].

Falling is a serious public health issue, especially among older adults. The convolutional neural network (CNN), which is a deep learning technology, has been identified as a useful AI technology that has the ability to predict sophisticated patient outcomes [27]. In 2020, research was conducted using CNN to predict the time of falling among Alzheimer’s patients, and it was found to be an optimal method for determining falling events that would assist in designing a customized approach based on the predicted time of fall [19]. Although AI-based technologies have been found to improve rehabilitation practices, there is still a need to investigate PTs’ knowledge, attitudes, and practices regarding AI in rehabilitation management. The main purpose of this study was to describe the current PTs’ understanding of AI applications in rehabilitation settings. The study also investigated the barriers that hold or delay PTs from adopting AI applications at work.

## 2. Materials and Methods

### 2.1. Participants

This study targeted PTs who are currently working in any academic or non-academic settings such as hospitals, clinics, home healthcare, or universities, and they were invited to participate voluntarily in the study. Participants had to be PT professionals to participate. The ethical approval for conducting this study was obtained from the NITTE Institute of Physiotherapy, NITTE University (NIPT/IEC/117/18/01/21).

### 2.2. Materials

#### 2.2.1. Validation of the Questionnaire

A 22-question questionnaire was developed and adapted from previous studies [12,13] and feedback from PTs who have experience in survey studies. Five PTs were contacted and invited to review the questionnaire questions. Each of the experts had to rate each question in the questionnaire for its clarity, structure, and relevance. Another round was performed after modifying the questions based on PTs experts’ feedback to establish the content validity of the survey questions. Eighty percent agreement on experts’ feedback was required for each question to finalize the questionnaire.

#### 2.2.2. Questionnaire Form

Questions 1 to 8 aimed to collect the demographic characteristics of the participants, including age, gender, PT license, experience years as PT, educational degree, primary workplace setting, and sub-specialty in PT. Questions 9 to 11 sought PT participants’ knowledge regarding AI-based technologies that are used in general, healthcare, and rehabilitation fields. Knowledge questions were captured using a yes/no format. In question 12, PTs were asked to select all the possible sources of AI information they mostly depend on. Question 13 asked PTs about the number of AI applications at work, and it was presented in a multiple-choice format from no application to more than 4. Questions 14 and 15 were designed to have participants’ attitudes toward the advantages of rehabilitation. Question 16 investigated the opinion of participants on the impact of AI technologies on the future of rehabilitation. The AI advantages, uses, and impacts question items were assessed using a 5-point Likert scale coded as 5 = strongly agree, 4 = agree, 3 = neutral, 2 = disagree, 1 = strongly disagree. Questions 17 and 18 investigated the ethical implications of AI technologies in rehabilitation, and it was snapshotted using multiple choice formats. Question 19 was to determine whether the PTs think that AI applications should be taught in rehabilitation curricula. Questions 20 and 21 were open-ended questions where PTs were asked to express and explain their response to the following questions:-In your opinion, which patients would benefit more from AI applications and why (musculoskeletal, geriatrics, neurologically impaired, etc.)? Please explain your response.-In your opinion, what are the major challenges or barriers that may limit AI applications?

Lastly, question 22 was to know how willing PTs are to receive more information on AI. Appendix A shows the questionnaire questions and their corresponding type of options.

In this study, PTs’ knowledge and attitudes were explained based on five predictors, including gender, years of experience, educational degree, subspeciality, and workplace. Regarding the variable categories, the years of experience variable was dichotomous (>10 years or ≤10 years). The workplace variable was categorized as academic or non-academic. Educational degree categories were undergraduate or postgraduate (Master’s and Ph.D.), while subspeciality categories were musculoskeletal, neurorehabilitation, and general.

### 2.3. Study Design

This study is a mixed-method design. In this study, the investigators embedded a qualitative component within a preliminary quantitative design to support the findings of the quantitative data to help an in-depth understanding of the research problem. The quantitative design element of the study used a cross-sectional, predictive design with exploratory predictors to understand the PTs’ knowledge and attitudes toward AI applications in rehabilitation. However, the qualitative part utilized open-ended questions, and it permitted the principal investigator (PI) to create themes of participants’ responses that would lead to the further discovery of PTs’ perceptions and acceptance of AI-based technologies in rehabilitation.

### 2.4. Procedure

The questionnaire was created using Google Forms (Google, LLC) in May 2021. A brief explanation of the study was posted in the preface section at the top part of the questionnaire with a highlighted statement that all information will be used confidentially and for the purpose of this study only. Informed consent was taken before answering the questionnaire to confirm the participation agreement.

PTs were recruited through e-mails and posts on social media (Facebook™, WhatsApp, and Twitter™). The snowball sampling method was facilitated by encouraging PTs to forward the electronic questionnaire to their colleagues in the PT sectors. The minimum sample size required to achieve a power of 0.8 was calculated using G-power software (latest ver. 3.1.9.7; Heinrich-Heine-Universität Düsseldorf, Düsseldorf, Germany). For a priori power calculation, a logistic regression test was chosen with an odd ratio of 1.5 and a significance level of 0.05. The minimum sample was indicated to be 208 respondents.

### 2.5. Statistical Analysis

Quantitative data were coded and then analyzed using IBM Statistical Package for Social Sciences software (SPSS), 26th edition, IBM, United States. All data were checked for completeness before the analyses. Continuous demographic data were analyzed using means and standard deviations, but in the case of categorical data, percentages and frequencies were used to describe the sample age, gender, years of experience, education qualifications, workplace settings, and the number of AI applications at work. Chi-square cross-tabulations and binary multivariate logistic regression tests were used to determine the predictors of AI knowledge and attitudes among PTs. The odds ratio (OR) and 95% confidence interval (CI) were reported to explain the relationship magnitude of the predictor variables with the dependent variable. For all the statistical analyses, α level of 0.05 or less was used to determine the statistically significant predictors.

For the qualitative analysis, open-ended questions were analyzed using thematic content analysis to interpret the meaning of the PT responses. The PI (M.A.) analyzed PTs’ responses using pre-established codes identified in the literature [28,29]. If any of the responses did not match the pre-established codes, the PI would generate a new code. The intercoder reliability (ICR) was established via peer review to assess the reliability of the coding [30,31]. The two peer reviewers were PTs, and they were asked to confirm the PI codes. In cases of disagreement or discrepancy between the two reviewers, a third member was asked to review and confirm the codes. The agreement percentage was above 80% for both questions. Once the coding phase was completed, similar codes were gathered into themes [29].

## 3. Results

### 3.1. Descriptive Statistics

Table 1 shows respondents’ sociodemographic characteristics. A total of two-hundred and thirty-six PTs from various subspeciality filled out the questionnaire. The respondents’ mean age was 35.20 with a standard deviation of ±6.97 years. The majority of the respondents (143, 59.6%) were males. A total of 52.5% (112) of the respondents had less than 10 years of experience in PT practice. A large number of the respondents (184, 61.87%) practice PT in non-academic settings such as hospitals, private clinics, and home health care. Neurorehabilitation and musculoskeletal were the most common subspecialty among PT respondents (37.1% and 27.1%, respectively). A total of 152 out of 236 (63.3%) reported the absence of AI applications at their working place. Only 12 (5%) respondents have come across more than four AI applications at work. However, social media was indicated to be the most used source for AI information among PTs by 29.83% of the total sample, followed by class lectures (25.18%) (Figure 1).

### 3.2. Factors Associated with AI Knowledge

#### 3.2.1. Simple Binary Logistic Regression

Table 2 shows the association among the predicted variables of AI knowledge among PTs. This study’s findings indicated that the employment sector, education qualification, years of experience, and specialty were significant predictors. For employment sectors, non-academic PTs were 1.77 times more likely to be knowledgeable in AI technologies than PTs who work in academic institutes (OR = 1.77 [95% CI: 1.01–3.12, *p* = 0.04]). Moreover, it was found that when holding less than ten years of experience constant, the odds of knowing about AI increased by 2.44 (95%CI: 1.40–4.22, *p* = 0.002) for PTs who have more than 10 years of experience. Education qualification was a significant predictor with an OR of 1.97 (95%CI: 1.11–3.50, *p* = 0.02). Compared to undergraduates, postgraduate PTs were 1.97 times more aware of AI-based technologies in PT clinical settings. Moreover, subspecialty was found to be a significant indicator of AI knowledge. Compared to the neurorehabilitation specialty, PTs who specialized in musculoskeletal or are general PTs were less likely to have knowledge about AI in rehabilitation by 0.52 (95%CI: 0.26–1.03, *p* = 0.06) and 0.36 times (95%CI: 0.19–0.70, *p* = 0.002), respectively. However, results showed that gender was not a significant predictor of AI knowledge among PTs (*p* = 0.76). Table 2 shows the detailed logistic regression analyses of the predicted variables of AI rehabilitation knowledge specific to PTs.

#### 3.2.2. Multivariate Logistic Regression Model

Multivariate logistic regression was performed to find the best predictors among different factors influencing knowledge of PTs toward AI uses in rehabilitation. In the 3-step model, the number of AI applications at work (OR = 3.81; 95% CI: 1.95–7.44]) was the best predictor, followed by experience (OR = 2.64; 95% CI: 1.46–4.79) and neurorehabilitation specialty (OR = 2.16; 95% CI: 1.03–4.45). Table 3 shows the association among different factors influencing the knowledge of PTs on AI use in rehabilitation established by multivariable logistic regression.

### 3.3. Factors Associated with AI Advantages

In order to have a snapshot of PTs’ attitudes towards AI-technology applications, respondents were asked to indicate their level of agreement towards three listed advantages of AI based on multiple predictors using a 5-point Likert scale. Respondents’ attitude toward AI is illustrated in Table 4.

#### 3.3.1. Reduce Professional Workload

In this study, 109 (46.2%) male PTs either agree or strongly agree that AI would reduce professional workload. It was also found that the same percentage of non-academic PTs agreed or highly agreed with this statement (109, 46.2%). Based on experience, the majority of PTs (94, 39.9%) who agreed or strongly agreed that AI would ease their workload were junior PTs with less than 10 years of experience. Regarding PTs’ education, the number of undergraduate PTs who either agreed or strongly agreed that AI could be used to reduce the load in clinical practices was lower in comparison to postgraduates (50 (21.2%) and 130 (55.1%)), respectively.

#### 3.3.2. Ease of Care

One hundred and thirteen (47.9%) male PTs reported their positive attitude towards implementing AI applications in clinical settings to facilitate patients’ ease of care. Moreover, ease of care was supported to be an advantage of AI by 50% (118) of the total sample size who work in the non-academic sector. Surprisingly, PTs in both categories (less or more than 10 years of experience) were almost equal in their level of positive agreement toward the ease of care statement (50 (40.2%) and 130 (40.1%)), respectively. The majority of respondents (138, 58.5%) from the undergraduate category had a positive opinion toward AI utilization in clinical practices to help in easing patient care.

#### 3.3.3. Diseases Prevention

The majority of male and female PTs (63 (26.7%) and 44 (18.6%)) reported positive attitudes toward the role of AI in disease prevention. However, 83 out of 263 (35.2%) respondents had no opinion toward this advantage based on gender. This study also found that a greater proportion of non-academic PT respondents (68, 28.8%) had a positive attitude toward using AI to prevent diseases. An almost equal percentage of junior and senior PTs believed that AI-based applications could be used to limit the burden of diseases (23.3% and 22.1%, respectively). Furthermore, 45.3% (107) of undergraduate and postgraduates expressed their greater agreement on disease prevention as an advantage of employing AI in rehabilitation practices.

### 3.4. Factors Associated with AI Uses

PT respondents reported their opinion regarding multiple AI uses in rehabilitation, and the opinions were captured using the Likert scale. Table 5 plots the detailed PTs’ attitudes toward AI uses in rehabilitation.

#### 3.4.1. Predicting Diseases

A great percentage of males (36%) had a positive opinion about the utilization of AI in rehabilitation to forecast patients’ medical status. The results also showed that non-academic respondents (89, 37.7%) were slightly higher in their agreement on employing AI-based technologies in rehabilitation to help therapists in predicting diseases than PT educators (55, 23.3%). Based on years of experience, this study found approximately equal agreement on using AI to generate disease prediction. A high percentage (101, 42.8%) of postgraduate PTs had a positive impression that AI facilitates disease prediction in rehabilitation settings.

#### 3.4.2. Goal Setting

Most of the male and female respondents (176, 74.6%) either agreed or strongly agreed that goal setting is a beneficial use of the AI system. Compared to academic PTs, a higher proportion of non-academic (49.2%) had positive attitudes toward AI as means to assist in developing goal settings based on patients’ health conditions. However, PTs had a nearly equal positive opinion that goal setting can be designed by AI. The majority of undergraduate and postgraduate PTs (176, 74.6%) believe that goal setting is a benefit of employing AI in clinical practices.

#### 3.4.3. Assistive Technologies

The study found male PTs (53.4%) significantly agreed or strongly agreed that AI is an assistive technology in rehabilitation. The findings also highlighted the positive attitude of non-academics (54.7%, 129) toward categorizing the use of AI applications as assistive technologies in patients’ management processes. Based on PTs’ experience and qualifications, only 4 out of 236 (1.6%) respondents had a negative opinion about using AI as an assistive technology tool in clinical practices.

#### 3.4.4. Diagnostic Tool

In this study, male PTs (99, 41.9%) had higher agreement than females (68, 29.2%) that AI applications are utilized to provide the diagnosis. The majority of non-academic PTs (107, 45.4%) stated their positive attitude toward AI as a diagnostic tool for several medical cases. Moreover, the results indicated that the majority of junior and senior PTs (168, 71.4%) highly believe in using AI for diagnostic determinations. However, using AI as a diagnostic tool was highly supported by postgraduate PTs (122, 51.7%).

### 3.5. Factors Associated with AI Impacts

Respondents were instructed to express their agreement level toward the three listed impacts. Detailed attitudes of PTs toward AI impacts are shown in Table 6.

#### 3.5.1. Reducing Human Resource

A total of 171 out of 236 (72.4%) male and female PTs either agreed or strongly agreed that AI has a role in reducing human resources. The findings also showed that non-academic PTs (100, 42.4%) supported that AI implementation would result in limiting human resources in rehabilitation. Only 20 respondents (8.5%) either disagreed or strongly disagreed that AI applications would have a negative impact on human resources. Fifty-three percent (134) of PTs who were Master’s or Ph.D. holders had a high level of agreement that AI would strike down the human workforce in clinical aspects.

#### 3.5.2. Increase Productivity

A high percentage of both genders had a positive opinion toward increasing productivity by AI in rehabilitation. Based on the workplace, non-academic PTs (115, 48.7%) positively supported the statement of increasing therapists’ work productivity by AI. Few PTs (6, 2.5%) had negative attitudes that AI would help to enhance work productivity in rehabilitation. It was found that 58.1% of the postgraduate PTs were significantly higher in their agreement than undergraduates (22.1%) that AI would be a facilitator for productivity in clinical practices.

#### 3.5.3. Improve Patient Quality of Life

The study found that male PTs (44.5%) were higher than females (29.2%) in their agreement that AI-based technologies have an impact on increasing patients’ quality of life. Additionally, the study results implied that non-academic PTs (114, 48.3%) were positive toward employing AI to improve patients’ quality of rehabilitation. Equal proportions of PTs totally agreed on the positive impacts of AI systems based on their years of experience. The current study results showed a significant positive agreement among postgraduate respondents that improving quality of life could be a result of implementing AI technologies in healthcare.

The last section of the questionnaire explored the opinions of PTs regarding the ethical implications of AI in rehabilitation. A total of 92 out of 236 (40.1%) PTs respondents expressed their primary fear of using AI applications as the inability of the AI technology to produce clinical reasonings for the cases beyond its programming scope. Moreover, 35.2% (83) of the respondents reported their ethical concerns about the AI system and their failure to understand or feel human beings. However, only 25.8% (61) of the respondents were worried that AI technology creators might have minimal or no experience in clinical practices.

A question was asked, “which decision should be taken if there was a conflict between AI and clinicians’ predictions?” The majority of PT respondents (177, 75.0%) believed the clinicians’ opinion should be considered in case of conflict, whereas 51 (21.61%) thought that patients’ preferences should be prioritized over AI and clinicians’ judgments. However, very few PTs respondents (3.4%) believed that AI produces trusted predictions. In this study, results showed that the majority of the respondents (186, 78.8%) believed that AI courses should be taught and integrated into the PT curriculum.

### 3.6. Qualitative Data Analysis

Upon reviewing the responses, it was clear that the majority of PTs have limited knowledge and skills to adopt AI-advanced technologies being used in clinical practices.

#### 3.6.1. The First Qualitative Question Analysis 

The first qualitative question was navigating PTs’ opinions regarding which patients would benefit more from AI-based applications; neurorehabilitation, geriatric, and musculoskeletal impairments were the most indicted patients’ conditions that could benefit from AI implementation in rehabilitation settings. Pre-established codes were driven by previous research [4,11,23,24,32,33,34,35,36,37]. Moreover, PTs were asked to explain the benefit of AI technologies in the rehabilitation of patients. Five themes were generated from PTs’ responses to explain their selections (Figure 2).

**Theme 1.** All patients based on the impairments.

The majority of PTs stated that all patients could gain advantages from AI but based on the disease or impairment. For example, one respondent said, “*AI gives benefits to patients depends on disease condition, impairment*.” Another respondent also said, “*All of them as AI is a tool to make things easy, and it can be developed and used based on individual rather than for a particular department*”.

On the other hand, a high number of the respondents thought that neurological and geriatrics patients would obtain the most benefit from AI-based technology. PT respondents explained their selection of these two specific conditions as AI would help those patients in their daily life activities, help therapists in their management process and assist in expecting patients’ responses.

*“Geriatric, neurologically impaired because it can assist these kinds of cases in their daily activities and predict their response.”*—Participant 163.

Moreover, a set of PT respondents believed that musculoskeletal and sport injury patients could obtain more advantages from AI applications in PT settings because most of the musculoskeletal cases are not cognitively impaired and that may allow them to follow the programmed instructions easily.

*“Musculoskeletal, as it will be easier for the patient to understand and apply.”*—Participant 179.

**Theme 2.** Selected patients based on AI advantages.

Several respondents believed that using AI technology would help to reduce therapists’ workload when managing selected cases such as musculoskeletal and neurological because patients’ movement can be monitored or guided by AI smart machines.

*“I think musculoskeletal and neuro patients would benefit more from AI compared to other areas because, by the application of AI, rehabilitation can be performed more precisely and accurately with a constant rhythm throughout the session when compared to manual techniques.”*—Participant 126.

The majority of the respondents also stated that AI could assist therapists in managing treatment sessions in the absence of human power, which may reduce their workload.

*“Geriatrics and neurological impaired because they will be guided by AI to do things correctly even in the absence of a Physiotherapist.”*—Participant 14.

**Theme 3.** Selected patients based on AI uses.

Many respondents mentioned various uses of AI in rehabilitation, such as monitoring or correcting patients’ movements during the therapeutic session and customizing treatment plans based on patients’ input data. For example, a respondent explained the use of AI as, *“Artificial limbs and rehabilitation, AI can play a major role in movement learning and error correction.”*—Participant 29. Another respondent wrote, *“Almost all the patient groups shall be benefited by AI, provided it is customized.”*—Participant 20.

Furthermore, many respondents mentioned that AI-based technologies could be used to provide feedback for some cases who need to be encouraged throughout the PT session, which helps to improve patient outcome measures.

*“Neurological impairments as most therapies targeting neurological disorders are feedback based… so the more accurate feedback the more accurate outcome.”*—Participant 30.

*“Definitely it will support clinicians’ effort to treat a neurologically impaired patient like visual or audible feedback is necessary to retrain the least amount of response from the patient.”*—Participant 192.

**Theme 4.** Selected patients based on AI impacts.

Some respondents mentioned the impact of AI-enabled applications in improving the quality of care via guiding PTs throughout the assessment and treatment process. Additionally, respondents revealed the importance of AI technologies to help therapists in monitoring patients’ progress and adherence to home programs.

*“Mostly all of the above mentioned will be benefited as would help them to work more efficiently, effectively and consistent way.”*—Participant 189.


*“It can help to those staying in remote areas where availability of medical facilities is less. As well it can help even a Physiotherapist to track record and keep data for analysis of progress.”*


**Theme 5.** Selected patients based on AI ethical and trust issues.

A few participants stated that they have no experience in utilizing any AI-based smart technologies in their practice, and that raises their ethical concerns and questions about the ability of AI to replace human efforts.

*“As I haven’t experienced the applications of AI in all sectors, I am not sure. Still, I think musculoskeletal patients may be benefitted as in Neurological cases are more complicated the judgement of therapist matters more.”*—Participant 96.

#### 3.6.2. The Second Qualitative Question Analysis

The second qualitative question asked about the perceived barriers that might limit AI utilization in rehabilitation. Respondents identified several barriers that may limit the usability of AI, in their opinion. The researchers started coding the data based on the pre-established codes and applied them to the data set. Codes were primarily derived from the literature [12,14,38,39,40,41]. Seven themes were derived from the response of the PTs. Figure 3 displays the frequency of the themes.

**Theme 1.** Inability of AI to manage all patients’ health conditions or impairments.

Very few participants were concerned about the inability of AI applications to be customized according to the patient’s conditions, impairments, or clinical scenario. *“Clinical situations to suit its application”*—Participant 1.

Some respondents also responded negatively about the capability of AI technologies to accommodate or handle different cases: *“Inability of AI to cater to a variety of patients”*—Participant 30.

**Theme 2.** Cost and available resources of AI in clinical settings.

Many respondents reported cost as the main barrier to AI implementation, particularly in PT practices. Respondents identified costs as the cost of AI machines and the cost of treatment. Two respondents wrote, “*All AI equipment are very costly and even the patient couldn’t afford the treatment fees*.” Another said, “*Mainly the cost that can’t afford for all type of patients*”.

Moreover, most PT respondents indicated that not only the cost of the equipment and treatment are barriers to using AI but also the cost of implementation, such as developing software and giving AI education courses, and training the users. Respondents believe that healthcare providers and AI developers both need the training to use AI effectively, which might be expensive. For instance, one of the responses was, “First and foremost will be the cost of implementation. Additionally, difficulties in procuring and developing the hardware and software for AI based rehab; Intense training needed for the operator as well as the service user; digital illiteracy of the therapists etc. are the major challenges”.

**Theme 3.** Compliance and adoption of AI among patients and therapists.

Most participants mentioned patients’ and/or therapists’ acceptance and adoption of AI technologies as major barriers to AI implementation. For example, one participant stated his opinion on barriers to implementing AI by saying, “*Acceptance by the patient and experienced professionals*”.

In addition, patient compliance was mentioned by the number of PT respondents as a concern of AI technologies in rehabilitation settings. One of the participants said, “*Investment cost is high as well as patient’s cost is also high which results in poor patient compliance*”.

**Theme 4**. Lack of knowledge and proficiency.

Various respondents admitted the insufficient knowledge and skills that they have regarding AI-based applications. One respondent expressed his opinion on AI barriers as “*lack of familiarity and knowledge about AI; clinical inertia to use such technology; cost and higher degree of skill required; availability of such applications; patient acceptance; inability to cater to a variety of patients*”.

In addition, respondents expressed that they are worried about AI developers being outside of the medical proficiency field and not having the clinical knowledge and skills that may affect the patient’s outcome. PTs respondents felt that collaboration among healthcare professionals and AI developers might lead to successful implementation.

*“Cost will obviously go towards higher side. And other limitation will be from developer side as they need to have the knowledge about medical profession, so in turn they will require to couple up with the medical professionals as whole team and need to careful design the script for its successful functionality and will need to set a perfect paradigm for its use.”*—Participant 54.

**Theme 5**. Technology trust in clinical settings.

Many participants expressed their trust issues toward AI applications that are used in clinical practices. Some respondents questioned the ability of AI to suit every clinical case scenario. Quality of care was also a major concern since PTs were not confident that AI could generate a customized plan of treatment for every patient. “*Every individual is different and requires tailor made protocol, which is not possible by AI. So, quality of treatment will be a major concern*.” In addition, some participants were worried about the automation function of AI, and they could not understand some of the complicated or advanced cases. “*Complex patient experiences may not be accurately captured by the software*.”—Participant 199.

Moreover, some PTs prefer traditional treatment given by humans as they have concerns about the technical issues that could arise from using machines: “*Being electronic device, malfunction may result in new problem to patients.*”—Participant 183.

**Theme 6.** Ethical implications of AI applications.

A number of the respondents mentioned therapist-patient interaction as a barrier to AI implementation in the medical field. PTs were questioning the automation function of AI and the absence of the human touch or emotion: “*Patients-therapist interaction is crucial in many cases management. That may be lacking in AI.*”—Participant 110.

Furthermore, some respondents were concerned about the absence of emotions, feelings, and human touch when using AI automation clinical tasks; some of the PTs think that empathy and human connection cannot be found in advanced management technology. For example, a respondent expressed his opinion as, “*In physical therapy human interaction and hands on therapies play a vital role and AI can’t completely take over that human factor. Also cost can be a barrier and reach may be limited to high end centers*”.

**Theme 7.** Patients’ perception and understanding of AI.

Patients’ perception, awareness, and understanding were also mentioned as barriers to AI application in rehabilitation: “*AI based rehab programs are generally costlier than the normal rehab. Also there is very less awareness among patients about it…*”.

One participant raised his concern about the understanding of the patients and the ability to follow the automation task. “*It is surely the patient’s perception on using AI applications as they may be not in a mental status or position to understand or follow the programmed treatment*”.

## 4. Discussion

The primary aim of this study was to understand PTs’ perceptions and identify the perceived factors that may limit AI adoption in rehabilitation. Moreover, the influence and association of multiple demographic factors on AI adoption were investigated. The key findings were that; years of experience, education qualification, subspeciality, and workplace were significant predictors of AI adoption among PTs. In this study, it was found that senior PTs, postgraduate, non-academic PTs, and neurorehabilitation specialties were significant indicators of knowledge of AI application in rehabilitation.

Although there are multiple benefits of AI in predicting patients’ diagnosis and prognosis, there is no clear evidence regarding the current understanding of PTs’ views and preparedness to use AI in their practices which raises the necessity for further exploration. This is the first study that discovers PTs’ knowledge and experience with AI-enabled applications in addition to the perceived barriers that may prevent therapists from operating AI in rehabilitation. Previous research believed that employing AI facilities would result in having a consistent plan of care that may increase clinical work productivity and quality of care [42]. That was similar to the results of this study which found some respondents supported the positive impact of AI-enabled applications on rehabilitation management. Nevertheless, a high number of respondents had a passive opinion regarding AI implementation due to the absence of AI applications at their work.

In clinical settings, practice is an important facilitator that helps in gaining the interest of therapists and clinicians to learn about AI and adopt it in their clinical applications. In this study, only 5% of the total sample reported their hands-on familiarity with AI applications at work. This was consistent with previous research that found that less than 10% of surgeons currently use robotic surgery techniques in hospitals, and 60% of surgeons documented the absence of AI and robotic technologies in their clinical practices [14]. The results emphasize the urgency of accelerating AI adoption and acceptance through training courses and workshops among PTs.

Among various healthcare specialties, the amount of healthcare providers’ experience was studied as a predictor of AI utilization in clinical practices. In this study, regression analyses showed that the more year of experience, the more likely to adopt AI applications among PTs. In New Zealand, a survey was targeted among medical physicists and radiation oncologists to describe the adoption of AI in their practices, and it was found that experience was positively associated with AI adoption [38]. A possible explanation might be that newly graduated have not learned about AI in their academic journey. However, that could be bridged by integrating AI into academic studies. In addition, gender was a significant factor that predicted the knowledge about AI tools among PTs. Researchers found that males reported having more knowledge and higher positive attitudes toward AI applications than females. This was consistent with a study that was conducted by Santos et al. [43], where males were more interested to know and learn about AI than females in medical schools. Moreover, Alsobhi et al. [44] found that experience and educational qualification were significant factors in knowledge about AI. This was in line with the findings of this study.

Subspeciality was a significant predictor of AI knowledge among respondents. Findings indicated that neurorehabilitation PTs were more knowledgeable than other PT subspecialties. There were no previous studies conducted to understand the PTs’ attitudes and perceptions about AI. Nevertheless, the embedded open-ended questions provided further understanding of the regression analysis, so many respondents thought that neurological patients need AI tools to improve their quality of life and functionality. Another possible explanation of the specialty factor was that many AI studies had been conducted in the field of neurology rehabilitation, such as stroke [3] and Parkinson’s [45].

The qualitative part allowed having an in-depth understanding of AI adoption and acceptance among providers. High numbers of PT respondents reported their concerns about patients’ perceptions and acceptance of automation and AI innovations in rehabilitation. In the literature, patient and healthcare system trust was documented as an essential factor in the successful implementation of AI in healthcare [15,16,17]. Moreover, researchers found that the general population exhibited AI-technology trust issues in its function, such as diagnosis and prediction of their health conditions [41]. As a result, a lack of trust should be perceived as a factor that limits the uptake of AI technologies in rehabilitation facilities.

On the other hand, many respondents expressed their worries about the absence of human touch when relying on AI tools, especially in rehabilitation. In addition, therapist-patient communication was mentioned as an ethical concern of AI utilization in healthcare. Previous studies had similar findings that using AI tools might eliminate the effective interactions between healthcare providers and their patients [46]. Moreover, a study was conducted to evaluate the robot’s functions to assist the elderly and their caregivers in monitoring safety and health status. Researchers found that robotic applications improved patients’ quality of life, but patients reported their concerns about the limited human interaction with AI tools [47]. The results of this study suggested addressing the potential communication barriers between AI-based devices and consumers and finding ways to make them more interactive.

This study identified a lack of knowledge as a barrier to adopting AI among therapists, although AI technologies have promises to facilitate health delivery systems by providing diagnosis and prognosis, which contribute to optimal care. In this study, 144 out of 236 PT respondents reported less knowledge about AI technologies in both healthcare and rehabilitation fields than in general AI applications. In rehabilitation research, AI is being applied in the form of computer interfaces, virtual reality, and exoskeleton rehabilitation programs [27,28,30]. However, the intention to use AI would be increased if the AI users enrich their AI knowledge. Similar findings were highlighted in a study that was conducted by Sun and Medaglia [41], who reported that lack of knowledge was a possible restriction of AI implementation.

### Study Strengths and Limitations

The strength of this study was employing the mixed-method design where the qualitative data supplement the primary quantitative information regarding PTs’ existing knowledge and opinion regarding AI, which provided an in-depth understanding of perceived barriers of limited AI utilization in rehabilitation. Another strength was investigating the associations between AI adoption and multiple associated factors. As with any research, some limitations could be addressed. Convenience sampling may limit the generalizability of this study’s findings. Moreover, the self-reported questionnaire may permit bias in some responses. The design of the study also did not allow for the investigation of the causal relationship.

The results of this study could be used as a baseline for further research to explore the adoption of AI clinical tools in rehabilitation. Future studies could investigate patients’ perceptions of AI applications to add to the results of this study. In addition, stakeholders could be interviewed in the future to investigate their preparedness to utilize AI in medical practices. The absence of AI understanding among the general public and healthcare professions should be considered a factor in translating AI-advanced research into medical practices.

## 5. Conclusions

The current study findings highlighted the limited knowledge and applications of AI in rehabilitation sectors. Moreover, the cost and available resources of AI were the most common barriers addressed by PTs. The study also demonstrated that AI application at the workplace, years of experience, and sub-specialty were associated with AI knowledge and attitudes among PTs. Future studies should focus on improving AI knowledge among PTs to bridge the gap between the existing research evidence and current PT practices.

## Figures and Tables

**Figure 1 ijerph-19-15919-f001:**
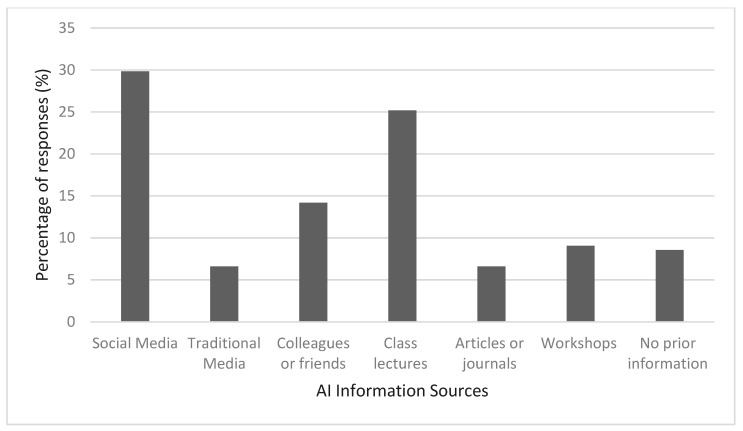
Multiple responses percentages of AI information sources. Social media and class lectures were the frequent documented source of AI knowledge among PTs.

**Figure 2 ijerph-19-15919-f002:**
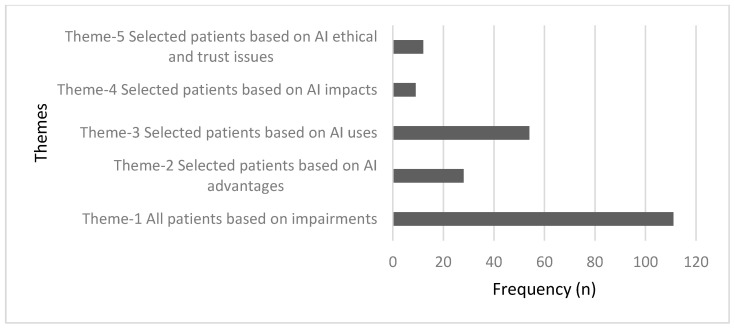
Frequency of the main themes generated from PTs’ responses to the first open-ended question. Theme 1 was the predominant theme that showed most PTs believed in the customization of AI based on patients’ impairments.

**Figure 3 ijerph-19-15919-f003:**
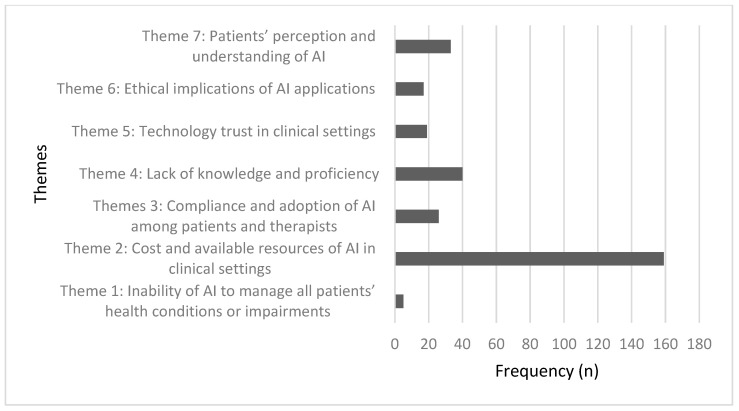
Frequency of the main themes generated from PTs’ responses to the second open-ended question. Cost and available resources of AI were the frequently documented barrier to AI adoption among PTs.

**Table 1 ijerph-19-15919-t001:** Sociodemographic variables of the participants (n = 236).

Variable		$\bar{\mathrm{mean}\;}$ ± SD	median (min–max)
**Age (Years)**		35.20 ± 6.97	35.50 (22–56)
		**Frequency (n)**	**Percentage (%)**
**Gender**	Male	143	(59.6)
	Female	93	(38.8)
**Work setting**	Academic	88	(36.7)
	Non-academic	148	(61.7)
**Educational qualification**	Undergraduate	69	(28.7)
	Postgraduate	167	(69.6)
**Subspecialty**	Cardiopulmonary	18	(7.5)
	General	43	(17.9)
	Geriatrics	2	(0.8)
	Musculoskeletal	67	(27.9)
	Neurorehabilitation	89	(37.1)
	Pediatrics rehabilitation	12	(5.0)
**Workplace AI applications**	0	152	(63.3)
	1	35	(14.6)
	2 to 4	37	(15.4)
	More than 4	12	(5.0)
**AI ethical implications**	Technology trust	92	(40.1)
	Empathy	83	(35.2)
	Users’ proficiency	61	(25.8)
**AI curriculum implementation**	Yes	186	(78.8)
	No	50	(21.2)
**AI knowledge**	General	211	(89.4)
	Healthcare	150	(75.4)
	Rehabilitation	178	(63.6)

Note. SD; standard deviation, min; minimum, max; maximum.

**Table 2 ijerph-19-15919-t002:** Logistic regression to assess the factors associated with AI.

	Variable	B	95% CI for B		SE B	β	*p* Value
			LL	UL			
**Knowledge about AI in rehabilitation**							
Gender	Constant	0.50			0.21		
	Male	0.08	0.63	1.87	0.28	1.09	0.76
	Female	Reference					
Employment Sector	Constant	0.36			0.17		
	Non academic	0.57	1.01	3.12	0.29	1.77	0.04
	Academic	Reference					
Experience	Constant	0.16			0.18		
	>10 years	0.89	1.40	4.22	0.28	2.44	0.002
	<10 years	Reference					
Qualification	Constant	0.09			0.24		
	Postgraduate	0.68	1.11	3.50	0.29	1.97	0.02
	Undergraduate	Reference					
AI in work place	Constant	0.13			0.16		
	1 or more AI in workplace	1.39	2.12	7.67	0.33	4.03	≤0.0001
	No AI in workplace	Reference					
Specialty	Constant	1.11			0.25		
	Musculoskeletal	−0.66	0.26	1.03	0.35	0.52.	0.06
	General	−1.01	0.19	0.70	0.33	0.36	0.002
	Neurorehabilitation	Reference					

**Table 3 ijerph-19-15919-t003:** Association between different factors influencing the knowledge of PTs on AI use in rehabilitation.

Variable	B	95%CI for B		SE B	β	% Predictability
		LL	UL			
**Step 1**						63.6
Constant	0.13			0.16		
**AI in workplace**						
1 or more AI in workplace	1.39	2.12	7.67	0.33	4.03 ***	
No AI in workplace	Reference					
**Step 2**						67.4
Constant	−0.23			0.21		
**AI in workplace**						
1 or more AI in workplace	1.36	2.04	7.51	0.33	3.91 ***	
No AI in workplace	Reference					
**Years of experience**						
<10 years	0.85	1.32	4.14	0.29	2.34 **	
>10 Years	Reference					
**Step 3**						72.0
Constant	−0.44			0.32		
**AI in workplace**						
1 or more AI in workplace	1.34	1.95	7.44	0.34	3.81 ***	
No AI in workplace	Reference					
**Years of experience**						
<10 years	0.97	1.46	4.79	0.30	2.64 ***	
>10 Years	Reference					
**Specialty**						
General	−0.28	0.37	1.54	0.36	0.76	
Neurorehabilitation	0.77	1.03	4.45	0.38	2.16 *	
Musculoskeletal	Reference					

*** *p* ≤ 0.001, ** *p* ≤ 0.01, * *p* ≤ 0.05.

**Table 4 ijerph-19-15919-t004:** Participant’s attitudes on AI advantages in clinical practices.

		Strongly Agree	Agree	Neutral	Disagree	Strongly Disagree	Total
**Reducing therapist workload**							
*Gender*	Male	38 (16.1)	71 (30.1)	31 (13.1)	2 (0.8)	1 (0.4)	143 (60.6)
	Female	29 (12.3)	42 (17.8)	21 (8.9)	1 (0.4)	0 (0)	93 (39.4)
*Employment sector*	Academic	25 (10.6)	46 (19.5)	17 (7.2)	0 (0)	0 (0)	88 (37.3)
	Non academic	42 (17.8)	67 (28.4)	35 (14.8)	3 (1.3)	1 (0.4)	148 (62.7)
*Experience*	>10 years	26 (11.0)	60 (25.4)	25 (10.6)	1 (0.4)	0 (0)	112 (47.5)
	<10 years	41 (17.4)	53 (22.5)	27 (11.4)	2 (0.8)	1 (0.4)	124 (52.5)
*Qualification*	Postgraduate	50 (21.2)	80 (33.9)	35 (14.8)	1 (0.4)	1 (0.4)	167 (70.8)
	Undergraduate	17 (7.2)	33 (14.0)	17 (7.2)	2 (0.8)	0 (0)	96 (29.2)
**Easing the patient care**							
*Gender*	Male	41 (17.4)	72 (30.5)	25 (10.6)	4 (1.7)	1 (0.4)	143 (60.6)
	Female	25 (10.6)	53 (22.5)	14 (5.9)	0 (0)	1 (0.4)	93 (39.4)
*Employment sector*	Academic	23 (9.7)	50 (21.2)	15 (6.4)	0 (0)	0 (0)	88 (37.3)
	Non academic	43 (18.2)	75 (31.8)	24 (10.2)	4 (1.7)	2 (0.8)	148 (62.7)
*Experience*	>10 years	29 (12.3)	66 (28.0)	13 (5.5)	3 (1.3)	1 (0.4)	112 (47.5)
	<10 years	37 (15.7)	59 (25.0)	26 (11.0)	1 (0.4)	1 (0.4)	124 (52.5)
*Qualification*	Postgraduate	47 (19.9)	91 (38.6)	25 (10.6)	3 (1.3)	1 (0.4)	167 (70.8)
	Undergraduate	19 (8.1)	34 (14.4)	14 (5.9)	1 (0.4)	1 (0.4)	69 (29.2)
**Prevention of diseases**							
*Gender*	Male	29 (12.3)	34 (14.4)	51 (21.6)	24 (10.2)	5 (2.1)	143 (60.6)
	Female	14 (5.9)	30 (12.7)	32 (13.6)	16 (6.8)	1 (0.4)	93 (39.4)
*Employment sector*	Academic	13 (5.5)	26 (11.0)	39 (16.5)	10 (4.2)	0 (0)	88 (37.3)
	Non academic	30 (12.7)	38 (16.1)	44 (18.6)	30 (12.7)	6 (2.5)	148 (62.7)
*Experience*	>10 years	16 (6.8)	36 (15.3)	42 (17.8)	16 (6.8)	2 (0.8)	112 (47.5)
	<10 years	27 (11.4)	28 (11.9)	41 (17.4)	24 (10.2)	4 (1.7)	124 (52.5)
*Qualification*	Postgraduate	26 (11.0)	50 (21.2)	63 (26.7)	24 (10.2)	4 (1.7)	167 (70.8)
	Undergraduate	17 (7.2)	14 (5.9)	20 (8.5)	16 (6.8)	2 (0.8)	69 (29.2)

**Table 5 ijerph-19-15919-t005:** Participants’ attitudes toward AI use in clinical practice.

		Strongly Agree	Agree	Neutral	Disagree	Strongly Disagree	Total
**Disease prediction**							
*Gender*	Male	26 (11.0)	59 (25.0)	47 (19.9)	10 (4.2)	1 (0.4)	143 (60.6)
	Female	14 (5.9)	45 (19.1)	25 (10.6)	8 (3.4)	1 (0.4)	93 (39.4)
*Employment sector*	Academic	18 (7.6)	37 (15.7)	26 (11.0)	6 (2.5)	1 (0.4)	88 (37.3)
	Non academic	22 (9.3)	67 (28.4)	46 (19.5)	12 (5.1)	1 (0.4)	148 (62.7)
*Experience*	>10 years	20 (8.5)	57 (24.2)	25 (10.6)	10 (4.2)	0 (0)	112 (47.5)
	<10 years	20 (8.5)	47 (19.9)	47 (19.9)	8 (3.4)	2 (0.8)	124 (52.5)
*Qualification*	Postgraduate	28 (11.9)	73 (30.9)	50 (21.2)	15 (6.4)	1 (0.4)	167 (70.8)
	Undergraduate	12 (5.1)	31 (13.1)	22 (9.3)	3 (1.3)	1 (0.4)	69 (29.2)
**Goal setting**							
*Gender*	Male	40 (16.9)	66 (28.0)	29 (12.3)	5 (2.1)	3 (1.3)	143 (60.6)
	Female	20 (8.5)	50 (21.2)	17 (7.2)	6 (2.5)	0 (0)	93 (39.4)
*Employment sector*	Academic	18 (7.6)	42 (17.8)	26 (11.0)	2 (0.8)	0 (0)	88 (37.3)
	Non academic	42 (17.8)	74 (31.4)	20 (8.5)	9 (3.8)	3 (1.3)	148 (62.7)
*Experience*	>10 years	27 (11.4)	62 (26.3)	22 (9.3)	1 (0.4)	0 (0)	112 (47.5)
	<10 years	33 (14.0)	54 (22.9)	24 (10.2)	10 (4.2)	3 (1.3)	124 (52.5)
*Qualification*	Postgraduate	40 (16.9)	79 (33.5)	38 (16.1)	8 (3.4)	2 (0.8)	167 (70.8)
	Undergraduate	20 (8.5)	37 (15.7)	8 (3.4)	3 (1.3)	1 (0.4)	69 (29.2)
**Assistive technologies**							
*Gender*	Male	50 (21.2)	76 (32.2)	15 (6.4)	2 (0.8)	0 (0)	143 (60.6)
	Female	38 (16.1)	44 (18.6)	11 (4.7)	0 (0)	0 (0)	93 (39.4)
*Employment sector*	Academic	30 (12.7)	49 (20.8)	9 (3.8)	0 (0)	0 (0)	88 (37.3)
	Non academic	58 (24.6)	71 (30.1)	17 (7.2)	2 (0.8)	0 (0)	148 (62.7)
*Experience*	>10 years	44 (18.6)	62 (26.3)	6 (2.5)	0 (0)	0 (0)	112 (47.5)
	<10 years	4 (18.6)	58 (24.6)	20 (8.5)	2 (0.8)	0 (0)	124 (52.5)
*Qualification*	Postgraduate	70 (29.7)	83 (35.2)	13 (5.5)	1 (0.4)	0 (0)	167 (70.8)
	Undergraduate	18 (7.7)	37 (15.7)	13 (5.5)	1 (0.4)	0 (0)	69 (29.2)
**Diagnostic tool**							
*Gender*	Male	43 (18.2)	56 (23.7)	36 (15.3)	4 (1.7)	4 (1.7)	143 (60.6)
	Female	26 (11.0)	42 (18.2)	20 (8.5)	4 (1.7)	0 (0)	93 (39.4)
*Employment sector*	Academic	24 (10.2)	37 (15.7)	23 (9.7)	4 (1.7)	0 (0)	88 (37.3)
	Non academic	45 (19.1)	62 (26.3)	33 (14.0)	4 (1.7)	4 (1.7)	148 (62.7)
*Experience*	>10 years	39 (16.5)	46 (19.7)	25 (10.6)	2 (0.8)	0 (0)	112 (47.5)
	<10 years	30 (12.7)	53 (22.5)	31 (13.1)	6 (2.5)	4 (1.7)	124 (52.5)
*Qualification*	Postgraduate	49 (20.8)	73 (30.9)	37(15.7)	6 (2.5)	2 (0.8)	167 (70.8)
	Undergraduate	20 (8.5)	26 (11.0)	19 (8.1)	2 (0.8)	2 (0.8)	69 (29.2)
**Education enhancement**							
*Gender*	Male	52 (22.0)	63 (26.7)	23 (9.7)	1 (1.3)	2 (0.8)	143 (60.6)
	Female	29 (12.3)	51 (21.6)	10 (4.2)	3 (1.3)	0 (0)	93 (39.4)
*Employment sector*	Academic	27 (11.4)	48 (20.3)	9 (3.8)	4 (1.7)	0 (0)	88 (37.3)
	Non academic	54 (22.9)	66 (28.0)	24 (10.2)	2 (0.8)	2 (0.8)	148 (62.7)
*Experience*	>10 years	37 (15.7)	58 (24.6)	16 (6.8)	1 (0.4)	0 (0)	112 (47.5)
	<10 years	44 (18.6)	56 (23.7)	17 (7.2)	5 (2.1)	2 (0.8)	124 (52.5)
*Qualification*	Postgraduate	57 (24.2)	78 (33.1)	26 (11.0)	6 (2.5)	0 (0)	167 (70.8)
	Undergraduate	24 (10.2)	36 (15.3)	7 (3.0)	0 (0)	2 (0.8)	69 (29.2)

**Table 6 ijerph-19-15919-t006:** Participants attitudes toward the impact of AI on rehabilitation field.

		Strongly Agree	Agree	Neutral	Disagree	Strongly Disagree	Total
**Reducing human resource**							
*Gender*	Male	26 (11.0)	64 (27.1)	37 (15.7)	12 (5.1)	4 (1.7)	143 (60.6)
	Female	33 (14.0)	48 (20.3)	8 (3.4)	3 (1.3)	1 (0.4)	93 (39.4)
*Employment sector*	Academic	27 (11.4)	44 (18.6)	13 (5.5)	4 (1.7)	0 (0)	88 (37.7)
	Non academic	32 (13.6)	68 (28.8)	32 (13.6)	11 (4.7)	5 (2.1)	148 (62.7)
*Experience*	>10 years	22 (9.3)	59 (25.0)	22 (9.3)	7 (3.0)	2 (0.8)	112 (47.5)
	<10 years	37 (15.7)	53 (22.5)	23 (9.7)	8 (3.4)	3 (1.3)	124 (52.5)
*Qualification*	Postgraduate	45 (19.1)	80 (33.9)	29 (12.3)	10 (4.9)	3 (1.3)	167 (70.8)
	Undergraduate	14 (5.9)	32 (13.6)	16 (6.8)	5 (2.1)	2 (0.8)	69 (29.2)
**Increase productivity**							
*Gender*	Male	49 (20.8)	60 (25.4)	29 (12.3)	4 (1.7)	1 (0.4)	143 (60.6)
	Female	25 (10.6)	55 (23.3)	12 (5.1)	1 (0.6)	0 (0)	93 (39.4)
*Employment sector*	Academic	26 (11.0)	48 (20.3)	14 (5.9)	0 (0)	0 (0)	88 (37.3)
	Non academic	48 (20.3)	67 (28.4)	27 (11.4)	5 (2.1)	1 (0.4)	148 (62.7)
*Experience*	>10 years	35 (14.8)	64 (27.1)	12 (5.1)	1 (0.4)	0 (0)	112 (47.5)
	<10 years	39 (16.5)	51 (21.6)	29 (12.3)	4 (1.7)	1 (0.4)	124 (52.5)
*Qualification*	Postgraduate	54 (22.9)	83 (35.2)	28 (11.9)	2 (0.8)	0 (0)	167 (70.8)
	Undergraduate	20 (8.5)	32 (13.6)	13 (5.5)	3 (1.3)	1 (0.4)	69 (29.2)
**Improve patient quality of life**							
*Gender*	Male	55 (23.3)	50 (21.2)	29 (12.3)	7 (3.0)	2 (0.8)	143 (60.6)
	Female	29 (12.3)	40 (16.9)	22 (9.3)	2 (0.8)	0 (0)	93 (39.4)
*Employment sector*	Academic	23 (9.7)	37 (15.7)	26 (11.0)	2 (0.8)	0 (0)	88 (37.3)
	Non academic	61 (25.8)	53 (22.5)	25 (10.6)	7 (3.0)	2 (0.8)	148 (62.7)
*Experience*	>10 years	35 (14.8)	50 (21.2)	24 (10.2)	3 (1.3)	0 (0)	112 (47.5)
	<10 years	49 (20.8)	40 (16.9)	27 (11.4)	6 (2.5)	2 (0.8)	124 (52.5)
*Qualification*	Postgraduate	59 (25.0)	67 (28.4)	36 (15.3)	4 (1.7)	1 (0.4)	167 (70.8)
	Undergraduate	25 (10.6)	23 (9.7)	15 (6.4)	5 (2.1)	1 (0.4)	69 (29.2)

## Data Availability

The de-identified datasets generated through this study can be provided by the corresponding author upon request.

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
