# Peer review of "Facilitators and Barriers of Artificial Intelligence Applications in Rehabilitation: A Mixed-Method Approach"

_ijerph, 2022, doi:10.3390/ijerph192315919_

Round 1

Reviewer 1 Report

The authors conducted a mixed-method analysis based on survey data to identify the facilitators and barriers in applications of artificial intelligence (AI) among physical therapists (PTs). Overall, the paper is well written and organized. My comments are provided below.  

1.       In the Introduction section, the authors reviewed a number of articles on AI applications in general healthcare and rehabilitation settings. While it provided values to have a literature review in a broader healthcare context, a large portion of the review in this paper was focused on AI applications in general healthcare settings instead of the study setting – i.e., rehabilitation/PT practice. Only the paragraph from line 76 to 85 presented three articles that were relevant to rehabilitation. I would suggest the authors to include more reviews focusing on the rehabilitation setting/PT practice.

2.       To follow up the comment above, it would be informative to present what/how AI techniques have been applied in the rehabilitation setting. In other words, how would AI help rehabilitation care delivery services? In the review of the three relevant articles, one used AI (informatics and robotics) to train patients and monitor progress, one used AI (a virtual PT system) to improve the balance and mobility of patients with Parkinson’s disease remotely, and one used AI (AI enabled mobile applications) to monitor and enhance patient adherence to therapeutic exercises in various musculoskeletal cases. Are there any other AI applications in PT practices? If yes, please include them here.

3.       In Section 2.2.2., the authors provided a description of the survey questions. Some questions are self-explanatory, such as the demographic characteristics of participants (Q1-8), whether AI should be taught in rehabilitation curricula (Q19). However, some other questions need to be further elaborate, including the questions regarding participants’ AI knowledge (Q9-11), attitudes (Q14, 15), opinions on the impact of AI (Q16), and ethical implications of AI (Q17, 18). Also, some clarification is needed for AI advantages, uses and impacts question items. Were those sub-questions under Q16?  I would suggest using a table to list all the questions and options for each question.  

4.       In Section 2.5, second paragraph, the author mentioned that the PI (M.A.) analyzed PTs’ responses using pre-established coded identified in the literature. However, I did not find any citations in Section 3.6, which presented the results of qualitative data analysis.

5.       I have some concerns about the regression analyses presented in Section 3 Results. The authors conducted two types of regression: (1) simple binary logistic regression, and (2) multivariate logistic regression. These regression analyses were conducted to find the predictors that influence PTs’ AI knowledge and attitude.

1)      I do not capture how PTs’ knowledge and attitude were measured. For PTs’ knowledge, although the authors mentioned in Section 2.2.2. that the knowledge questions were captured using yes/no format, without knowing what those questions are, it is somewhat difficult to follow on the analyses. In addition, there were three knowledge questions (Q9-11), did the authors combined all three answers into one measure for PTs’ knowledge? The similar issues were present for the attitude measure.

2)      It is not clear to me why both simple and multivariate regression were needed? Further, why the authors used 3-step model in the multivariate regression? How were those additive factors (i.e., years of experience, specialty) selected? Were those selected based on the significance of individual independent variables derived from the simple regression? Please provide some justification for both regression analyses.

3)      In Section 3.2.2., the authors mentioned that “Multivariate logistic regression was performed to find the best predictors among different factors influencing knowledge and attitudes of PTs toward AI uses in rehabilitation. However, the caption of Table 3 only mentioned PTs’ knowledge. What about the results for PTs’ attitude?

6.       In Section 3.5.3., line 342, it says “However, very few PTs respondents (82.6%) believed that AI produces trusted predictions. The wording “very few” and the percentage “82.6” seem conflicting to each other.  

7.       In Section 4 Discussion, line 573, it says “This study identified lack of knowledge as a barrier to adopt AI among therapist …” Please provide some quantitative evidence based on the survey data. For Table 1, I would suggest including the descriptive analysis results for all the measurable variables, including knowledge, attitude, ethnical, whether AI applications should be taught.

8.       Some typos need to be fixed.

1)      Title: … : A mixed-method Approach. Please use upper case for the first letter of each word: A Mixed-Method Approach.

2)      Line 83, … by Lo et. Al., should be Lo et al.

3)      Line 166, …but in the case of categorical, data percentages…, should be but in the case of categorical data, percentages… (remove the comma after the word data).

Author Response

Responses to the reviewers' comments

Title: Facilitators and Barriers of Artificial Intelligence Applications in Rehabilitation: A Mixed-Method Approach

We would like to thank the reviewers for their comments and evaluations. The feedback helps us to improve the quality of this work.

Please find our responses to each comment below.

Reviewer 1:

The authors conducted a mixed-method analysis based on survey data to identify the facilitators and barriers in applications of artificial intelligence (AI) among physical therapists (PTs). Overall, the paper is well written and organized. My comments are provided below.  

Comment 1 & 2:

  1. In the Introduction section, the authors reviewed a number of articles on AI applications in general healthcare and rehabilitation settings. While it provided values to have a literature review in a broader healthcare context, a large portion of the review in this paper was focused on AI applications in general healthcare settings instead of the study setting – i.e., rehabilitation/PT practice. Only the paragraph from line 76 to 85 presented three articles that were relevant to rehabilitation. I would suggest the authors to include more reviews focusing on the rehabilitation setting/PT practice.

  1. To follow up the comment above, it would be informative to present what/how AI techniques have been applied in the rehabilitation setting. In other words, how would AI help rehabilitation care delivery services? In the review of the three relevant articles, one used AI (informatics and robotics) to train patients and monitor progress, one used AI (a virtual PT system) to improve the balance and mobility of patients with Parkinson’s disease remotely, and one used AI (AI enabled mobile applications) to monitor and enhance patient adherence to therapeutic exercises in various musculoskeletal cases. Are there any other AI applications in PT practices? If yes, please include them here.

Response to comments 1 and 2:

Initially, we did not focus on mentioning AI applications in PT practices because our goal was to explore and understand PTs’ opinions and barriers that may limit AI utilization among PTs. We agree with the reviewer adding more AI applications specifically in PTs will enhance the literature. So, we added to the literature recent studies that showed the importance of AI in PT rehabilitation. We added “Moreover, supervised machine learning was studied to investigate the ability of AI-enabled technology to monitor patients’ exercise adherence at home. A study done in 2018 by Burns et al. [25] demonstrated the technical feasibility of supervised machine learning to track adherence to rotator cuff exercise regimens at home among healthy individuals which improves patients’ healthcare outcomes. AI interventions have been developed not only as cost-effective procedures but also to enhance the quality of care. Researchers compared conventional and AI digital sessions after total knee replacement (TKR) surgery among patients with knee osteoarthritis. The digital sessions employed 3D movement quantification to detect patients’ motion via a phone application and a web-based site. The study concluded that digital intervention for home program after TKR surgery reduced the therapists’ workload and maximized patients’ outcomes [26]. Falling is a serious public health issue, especially among older adults. The convolutional neural network (CNN), which is a deep learning technology, has been identified as a useful AI technology that has the ability to predict sophisticated patients outcomes [27]. In 2020, research was done using CNN to predict the time of falling among Alzheimer’s patients, and it was found to be an optimal method for determining falling events that would assist in designing a customized approach based on the predicted time of fall [19].” In line 91 - 108.

Comment: 3

       In Section 2.2.2., the authors provided a description of the survey questions. Some questions are self-explanatory, such as the demographic characteristics of participants (Q1-8), whether AI should be taught in rehabilitation curricula (Q19). However, some other questions need to be further elaborate, including the questions regarding participants’ AI knowledge (Q9-11), attitudes (Q14, 15), opinions on the impact of AI (Q16), and ethical implications of AI (Q17, 18). Also, some clarification is needed for AI advantages, uses and impacts question items. Were those sub-questions under Q16?  I would suggest using a table to list all the questions and options for each question.

Response to comment 3:

We agreed with the reviewer’s suggestions that might be vague for the readers, so we listed all the questions in the questionnaire in a table. We referred to the questions in the questionnaire as Appendix A at the end of the manuscript.  “Appendix A shows the questions of the questionnaire and their corresponding type of options” in the text in section 2.2.2 line 158. The appendix was inserted at the end of the manuscript before the reference section.

Comment 4:

In Section 2.5, second paragraph, the author mentioned that the PI (M.A.) analyzed PTs’ responses using pre-established coded identified in the literature. However, I did not find any citations in Section 3.6, which presented the results of qualitative data analysis.

Response to comment 4: The method of analysis chosen for this study was a qualitative approach of thematic analysis. Deductive approaches use pre-existing research or knowledge-driven focus to identify themes of interest, we added the citation that guided us to apply the steps of identifying the pre-established code and analyzing the data in lines 208 and 215 section 2.5. Researchers drove the codes from previous research for the qualitative questions, and we added the citations for those studies in section 3.6 in lines 414-415 and 485-486.

Comment 5:

      I have some concerns about the regression analyses presented in Section 3 Results. The authors conducted two types of regression: (1) simple binary logistic regression, and (2) multivariate logistic regression. These regression analyses were conducted to find the predictors that influence PTs’ AI knowledge and attitude.

Comment 5.1:

I do not capture how PTs’ knowledge and attitude were measured. For PTs’ knowledge, although the authors mentioned in Section 2.2.2. that the knowledge questions were captured using yes/no format, without knowing what those questions are, it is somewhat difficult to follow on the analyses. In addition, there were three knowledge questions (Q9-11), did the authors combined all three answers into one measure for PTs’ knowledge? The similar issues were present for the attitude measure.

Response to comment 5.1:

We really apologize for the confusion in the questions and regression analysis; for a better understanding of the questions, the authors have attached the questions of the survey questionnaire with the option type in the appendix. For the regression analysis, the authors have taken into account only question number 11 “ Have you ever heard about any AI technologies used in rehabilitation” which assesses the knowledge of PT’s on AI. And for the multivariate analysis using multiple logistic regression also the authors have used question 11 mentioned above as the dependent variable and its option “YES” as the reference for the whole model.

Comment 5.2:

It is not clear to me why both simple and multivariate regression were needed? Further, why the authors used 3-step model in the multivariate regression? How were those additive factors (i.e., years of experience, specialty) selected? Were those selected based on the significance of individual independent variables derived from the simple regression? Please provide some justification for both regression analyses.

Response to comment 5.2:

 The authors appreciate the comments of the reviewer, here the authors have used the simple binary logistic regression to find the significant factors associated with the knowledge about AI in rehabilitation. Further, multivariate binary logistic regression was performed to see which among those factors are the best predictors for the knowledge of AI. In multivariate logistic regression a forward step-wise method was adopted in which the software automatically gives a 3 step model stating AI use in work place as the best predictor followed by years of experience and specialty. Here for the multivariate logistic regression, those factors which were significant in the simple logistic regression were added for analysis.  

Comment 5.3:

In Section 3.2.2., the authors mentioned that “Multivariate logistic regression was performed to find the best predictors among different factors influencing knowledge and attitudes of PTs toward AI uses in rehabilitation. However, the caption of Table 3 only mentioned PTs’ knowledge. What about the results for PTs’ attitude?

Response to comment 5.3:

The multivariate logistic regression was performed to find the best predictors among different factors influencing knowledge. We agree that the general statement of the purpose of the study to explore knowledge and attitudes give the anticipation of including the attitude in the multiple regression model, so we deleted the word “attitude” from the statement in line 273.

Comment 6:

In Section 3.5.3., line 342, it says “However, very few PTs respondents (82.6%) believed that AI produces trusted predictions. The wording “very few” and the percentage “82.6” seem conflicting to each other.  

Response to comment 6:

We appreciate the reviewer’s comment, and we apologize for the typo errors. The number of PTs who believed that AI produces trusted predictions was 8 respondents only which was equal to 3.4% (line 405).

Comment 7:

In Section 4 Discussion, line 573, it says “This study identified lack of knowledge as a barrier to adopt AI among therapist …” Please provide some quantitative evidence based on the survey data. For Table 1, I would suggest including the descriptive analysis results for all the measurable variables, including knowledge, attitude, ethnical, whether AI applications should be taught.

Response to comment 7:

We added the frequency of PTs who reported less knowledge in AI applications specifically in the healthcare and rehabilitation field in lines 640-642. We added “In this study, 144 out of 236 PT respondents reported less knowledge about AI technologies in both healthcare and rehabilitation fields than in general AI applications”. Also, we added the descriptive statistics for the measurable variables (knowledge, ethnical implication, and AI applications should be taught) in Table 1.

Comment 8:

Some typos need to be fixed.

8.1: Title: … : A mixed-method Approach. Please use upper case for the first letter of each word: A Mixed-Method Approach.

Response to comment 8.1: Edited as recommended

8.2: Line 83, … by Lo et. Al., should be Lo et al.

Response to comment 8.2: Edited as recommended

8.3: Line 166, …but in the case of categorical, data percentages…, should be but in the case of categorical data, percentages… (remove the comma after the word data).

Response to comment 8.3: Edited as recommended

Reviewer 2 Report

The authors are providing Physical Therapists’ views on Artificial Intelligence and identifying the barriers to use of AI in rehabilitation. Although they are doing a good job of explaining the literature review, methods, and results but the question here is that: How is this work any different from:

Alsobhi M, Khan F, Chevidikunnan M, Basuodan R, Shawli L, Neamatallah Z. Physical Therapists’ Knowledge and Attitudes 691 Regarding Artificial Intelligence Applications in Health Care and Rehabilitation: Cross-sectional Study. 2022. 24(10):e39565.

Unless the authors show another aspect or contribution of this research, there are not presenting anything new here. They need to clearly specify what is the contribution here and what is the novel work and what is being conducted that is different from the previous work.

Author Response

Reviewer 2:

Comment 1:

The authors are providing Physical Therapists’ views on Artificial Intelligence and identifying the barriers to use of AI in rehabilitation. Although they are doing a good job of explaining the literature review, methods, and results but the question here is that: How is this work any different from:

Alsobhi M, Khan F, Chevidikunnan M, Basuodan R, Shawli L, Neamatallah Z. Physical Therapists’ Knowledge and Attitudes Regarding Artificial Intelligence Applications in Health Care and Rehabilitation: Cross-sectional Study. 2022. 24(10):e39565.

Unless the authors show another aspect or contribution of this research, there are not presenting anything new here. They need to clearly specify what is the contribution here and what is the novel work and what is being conducted that is different from the previous work.

Response to comment 1:

The authors appreciate the comment of the reviewers, the article mentioned by the reviewer was published by the same authors of this manuscript. However, there are many differences between both of the manuscripts.

  1. The main objective of the current study was to find the barriers for the implementation of AI in rehabilitation, which was studied through a qualitative research methodology. And thus the design of the whole study became a Mixed model design including a qualitative and quantitative approach. Rather the previous study was just a quantitative study exploring the knowledge and attitudes of physical therapists regarding artificial intelligence.
  2. The current study was performed in India and the previous one was performed in Saudi Arabia
  3. The results of both studies are different in factors influencing knowledge.
  4. In the current study the authors used content analysis for the analysis of qualitative data.

Reviewer 3 Report

- To claim that PT worker lack sufficient exposure to AI, It would be helpful to include specific AI applications already implemented in PT somewhere. The literature is always ahead of the industrial and commercial products. Thus, PT worker are not aware of AI, maybe because there are not many real-life really available applications/products. For example, although many claim that quantum computers are the future, people have little exposure to quantum computers because there is none commercially available. 

- In which country was the survey conducted? Is the language different from the ones in references 12,13? Does the IRB approval apply to that country if different than the granting country?

- In general, the paper is well-written.

- Line 157, power of .8 --> 0.8

- typo on line 18, "Concurrent" --> "concurrent".

- Line 22-24, the statement can be stated differently to indicated, which group had the most or least AI knowledge.

Author Response

Reviewer 3:

Comment 1:

To claim that PT worker lack sufficient exposure to AI, It would be helpful to include specific AI applications already implemented in PT somewhere. The literature is always ahead of the industrial and commercial products. Thus, PT worker are not aware of AI, maybe because there are not many real-life really available applications/products. For example, although many claim that quantum computers are the future, people have little exposure to quantum computers because there is none commercially available.

Response to comment 1:

We agree with the reviewer adding more AI applications specifically in PTs will enhance the literature. So, we added to the literature with recent studies that showed the importance of AI in PT rehabilitation. We added “Moreover, supervised machine learning was studied to investigate the ability of AI-enabled technology to monitor patients’ exercise adherence at home. A study was done in 2018 by Burns et al. [25] demonstrated the technical feasibility of supervised machine learning to track adherence of rotator cuff exercise regimens at home among healthy individuals which improves patients’ healthcare outcomes. AI interventions have been developed not only as cost-effective procedures but also to enhance quality of care. Researchers compared conventional and AI digital sessions after total knee replacement (TKR) surgery among patients with knee osteoarthritis. The digital sessions employed 3D movement quantification to detect patients’ motion via a phone application and web-based site. The study concluded that digital intervention for home program after TKR surgery reduced the therapists’ workload and maximized patients’ outcomes [26]. Falling is a serious public health issue, especially among older adults. The convolutional neural network (CNN), which is a deep learning technology, has been identified as a useful AI technology that has the ability to predict sophisticated patients’ outcomes [27]. In 2020, research was done using CNN to predict time of falling among Alzheimer’s patients, and it was found to be an optimal method for determining falling events that would assist in designing customized approach based on the predicted time of fall [19].” In line 88

Comment 2:

In which country was the survey conducted? Is the language different from the ones in references 12,13? Does the IRB approval apply to that country if different than the granting country?

Response to comment 2:

The study was performed in India targeting physical therapists. Some survey questions were adapted from previous studies, and those questionnaires were written in English. The IRB was obtained from the county where the study was conducted while the fund was granted country was Saudi Arabia which has been mentioned in the acknowledgements.  

Comment 3:

In general, the paper is well-written.

Response to comment 3:

Thank you for your compliment

Comment 4:

Line 157, power of .8 --> 0.8:

Response to comment 4:

Edited as recommended

Comment 5:

typo on line 18, "Concurrent" --> "concurrent".

Response to comment 5:

Edited as recommended

Comment 6:

Line 22-24, the statement can be stated differently to indicate, which group had the most or least AI knowledge.

Response to comment 6:

We agree with the reviewer’s feedback and it was edited as “The major factors predicting a higher level of AI knowledge among PTs were being a non-academic worker (OR= 1.77 [95% CI; 1.01 to 3.12], pâ‚Œ0.04), being a senior PT (OR = 2.44, [95%CI: 1.40 to 4.22], pâ‚Œ 0.002), having a Master/Doctorate degree (OR= 1.97, [95%CI: 1.11 to 3.50], pâ‚Œ 0.02).”

Round 2

Reviewer 2 Report

The manuscript is accepted in the current form.